# Impacts of Semiochemical Traps Designed for *Bruchus rufimanus* Boheman 1833 (Coleoptera: Chrysomelidae) on Nontarget Beneficial Entomofauna in Field Bean Crops

**DOI:** 10.3390/insects14020153

**Published:** 2023-02-02

**Authors:** Arnaud Segers, Grégoire Noël, Louise Delanglez, Rudy Caparros Megido, Frédéric Francis

**Affiliations:** Functional and Evolutionary Entomology, University of Liège–Gembloux Agro-Bio Tech, Passage des Déportés, 2, 5030 Gembloux, Belgium

**Keywords:** faba bean, *Vicia faba*, bruchids, community structure, Apoidea, Syrphidae, Coccinellidae

## Abstract

**Simple Summary:**

The cultivation of seed legumes provides many services to biodiversity, the environment and agronomic systems. However, the field yields of these crops remain irregular and uncertain due to biotic and abiotic stresses. The introduction of crops, such as faba beans, must allow sufficient remuneration to farmers to meet sustainable development criteria. In this sense, it is essential to implement new strategies to control *Bruchus rufimanus* Boheman 1833 (Coleoptera: Chrysomelidae), the main pest of faba bean seeds. Recent discoveries in the chemical ecology of the pest led to the set-up of semiochemical traps opening the door to more sustainable biocontrol strategies. Here, we evaluate the efficacy of these traps for the capture of *B. rufimanus* by considering the phenology of the crop and the collateral effects of traps on beneficial insects.

**Abstract:**

Broad bean weevils (BBWs–Coleoptera: Chrysomelidae) are serious pests of field bean seeds that hamper the promotion of this crop in the diversification of European cropping systems. Recent research has identified different semiochemical lures and trap devices for the development of semiochemical-based control strategies of BBWs. In this study, two field trials were carried out in order to provide necessary information supporting the implementation of sustainable field use of semiochemical traps against BBWs. More particularly, three principal objectives were followed including (i) the identification of the most efficient traps for BBWs capture and the influence of trapping modality on BBWs sex-ratio, (ii) the assessment of eventual collateral effects on crop benefits including aphidophagous and pollinator insects such as Apidae, Syrphidae and Coccinellidae, (iii) the assessment of the crop developmental stage influence on the capture by semiochemical traps. Three different semiochemical lures were tested in combination with two trapping devices across two field trials in early and late flowering field bean crops. The crop phenology and climate parameters were integrated into the analyses to interpret the spatiotemporal evolution of the captured insect populations. A total of 1380 BBWs and 1424 beneficials were captured. White pan traps combined with floral kairomones were the most efficient traps for the capture of BBWs. We demonstrated that the crop phenology (c.f., the flowering stage) exerted strong competition on the attractiveness of semiochemical traps. Community analysis revealed that only one species of BBWs was captured in field bean crops (i.e., *Bruchus rufimanus*), and no trend was highlighted concerning the sex ratios according to the trapping devices. The beneficial insect community included 67 different species belonging to bees, hoverflies and ladybeetles. Semiochemical traps manifested a strong impact on beneficial insect communities that included some species under extinction threats and need to be further adapted to minimize such collateral effects. Based on these results, recommendations are provided for the implementation of the most sustainable BBWs control method that minimizes the impact on the recruitment of beneficial insects, which is an important ecosystem service for faba bean crops.

## 1. Introduction

One of the biggest challenges of current agriculture is to ensure food security while reducing its impact on the environment at the same time [1,2]. More sustainable cropping systems can be implemented at the national level by increasing crop diversity and promoting beneficial synergies between neighboring/successive crops in rotations [3,4]. In this sense, the introduction of legume crops such as *Vicia faba* L. (Fabaceae), commonly named field bean, faba bean or horse bean, is expected to greatly contribute to the sustainability of European feed/food production [5]. This culture provides many benefits to agroecosystems [6]. First, the plants have root symbioses with *Rhizobium* bacteria (i.e., nodules) that fix atmospheric nitrogen (N) and transform it into an uptaking form for plants. This natural green manure provides N fertilizers to the subsequent crop [7] and decreases greenhouse gases emission related to the production, transport and spreading, of synthetic fertilizers [8,9]. The main advantage of cropping field beans is the production of seeds with high starch or protein contents that can be used as feed or food. This contributes greatly to plant-protein autonomy in Europe and may replace unsustainable soybean imports from South America [10,11,12]. Finally, cropping field bean crops increases the abundance and the diversity of beneficial arthropods (i.e., biocontrol agents and pollinators) by providing a large amount of pollen and nectar [13]. Pollinators such as bumblebees, honeybees and wild bees feed on these floral resources. Predators of aphids such as lady beetles (Coleoptera: Coccinelidae), hoverflies (Diptera: Syrphidae), or some parasitoid species may also feed on the nectar provided by extrafloral nectaries [14,15,16,17].

The promotion of field bean introduction should, however, be economically reliable to meet sustainable development criteria. Yet, field bean crops show irregularities in seed quality and quantity due to biotic stresses such as root or leave diseases and pests [18,19,20]. One of the most important groups of pests for the valorization of seeds in the human food market is named broad bean weevils (BBWs). This group of pests corresponds to small Coleoptera belonging to the Chrysomelidae family (sub-family Bruchinae) and greatly impact the field bean seed quality. Five species (closely related morphologically and difficult to distinguish) were recorded in Belgium including *Bruchus rufimanus* Boheman 1833 (the most common one in field bean crops), *B. affinis* Frölich 1799, *B. atomarius* (L. 1761), *B. brachialis* Fåhraeus 1839 and *B. pisorum* (L. 1758) [21,22]. The pest stage of BBWs is the larvae developing inside forming seeds [23]. Quantitative damage consists in a loss of seed dry mass that may range from 5 to 9.4% [24,25,26] and qualitative damage includes a decrease in the seed germination capacity [27], a fall in nutritional and tasty properties caused by the accumulation of waste and feces by feeding larvae [28], and a decrease in the aesthetic quality caused by the perforation of emerging adults [29]. These injured seeds are then more susceptible to the development of phytopathogenic fungi in storage commodities such as *Penicillium* spp. or *Aspergillus* spp. [30,31]. Seed batches presenting infestation rates (i.e., the proportion of seeds presenting traces of post-embryonic development of BBW) higher than 2–3% are rejected from the food market [32,33], which is the more profitable outlet for growers exceeding by 30 euros/t the prices from feed markets [34,35].

The univoltine life cycle of BBWs starts in spring. Overwintering adults colonize crops at the blooming stage when temperatures exceed 15 °C (i.e., the activity threshold of flying adults). These adults are in reproductive diapause and need to feed on the nectar and pollen of host plants for sexual maturation [36]. Mating and oviposition are favored by temperatures exceeding 20 °C. Gravid females lay eggs on young pods and hatching larvae complete the whole post-embryonic development inside a single forming seed by endosperm consumption [37]. The emergence of the diapausing adults occurs either at the end of summer when mature seeds are harvested, i.e., in storage commodities or in the field depending on the timing of harvest [24], or the next spring when infested seeds are sown [38]. Around fifty percent of adults are reported to emerge during the harvesting period [39]. It should be noted that no secondary infestation impacts stored seeds as females are (i) in reproductive diapause and (ii) unable to lay eggs on dry seeds [40]. 

The control of BBWs is difficult because the damaging stage is out of reach of any phytosanitary intervention and because the adult’s behavior in crops is closely related to phenological and climatic factors [29,41,42]. European conventional methods of control consist of phytosanitary interventions with pyrethroids targeting adults before oviposition [33,43]. However, this method of control has shown increasing inefficiency over the last decades [41,44] and led to increasingly massive infestations since 2013. In consequence, the export market of seeds to Egypt for food uses collapsed [45] which has considerably decreased the field bean popularity [39,41].

New alternatives were recently suggested on the basis of chemical processes underlying the attraction of BBWs to the host plants. The volatile organic compounds (VOCs) emitted by faba bean flowers that attract adults searching for food resources (i.e., flower kairomone) were identified as a blend of (R)-limonene, (E)-ocimene, (R)-linalool, 4-allylanisole (i.e., estragol), cinnamyl alcohol, cinnamaldehyde, α and β-caryophyllene [32]. Three of these compounds were demonstrated to be effective for BBWs attraction in field experiments, i.e., (R)-linalool (17.7 mg/day), cinnamyl alcohol (0.4 mg/day) and cinnamaldehyde (0.77 mg/day). Later studies have identified another kairomonal signal emitted by pods having an attractive effect on gravid females searching for an oviposition site as a blend of cis-3-hexenyl acetate (30–40%), ocimene (15–20%), linalool (10–20%), α and β caryophyllene (10–20%), and limonene (15–20%) [46,47]. No pheromone was found to be effective in the field [29]. Semiochemical lures based on flower and pod kairomones were produced to develop semiochemical control strategies against BBWs, including monitoring and mass trapping strategies [41,48]. Two designs of traps can be used with these lures, a white pan trap with a transparent cylinder and a green funnel trap with barrier crossbars. Information about the influence of lures and trap design on the capture of BBWs is lacking while they play a strong influence on the trapping efficacy according to the insect’s behavior [49,50]. No information is provided on the influence of crop developmental stages (which may reduce the attractiveness of semiochemical traps), or the potential impact of these semiochemical traps on beneficial insect communities.

In this study, two field trials were carried out in winter field beans and spring field beans with four principal objectives: (i) comparing the effectiveness of different semiochemical traps (two trap designs and three semiochemical lures) to identify the best-suited trap for the control of BBWs population, (ii) assessing the influence of semiochemical trap design on BBWs sex-ratios, (iii) determining whether the attractiveness of semiochemical traps reproducing the crop odor were not in competition with the crop itself and (iv) characterizing the potential collateral effect of semiochemical traps on beneficial insects. This study is the first integrative and comparative semiochemical trapping trial in field bean crops. 

## 2. Materials and Methods

### 2.1. Plant Material

Two field trials were carried out during the cropping season 2020–2021 for the comparison of trapping modalities. Two cultures of field bean were cropped in Gembloux (Belgium) at geographical coordinates 50°50′34″ N, 4°73′19″ E and an altitude of 174.75 m. This area was known to have high BBWs populations according to previous studies [51], i.e., field beans were cropped under hyper-infestation conditions for the optimal discrimination of trap efficiency. These two crops included a winter field bean variety (WFB) named “Nebraska” and a spring field bean variety (SFB) named “Fanfare”. WFB and SFB were sown, respectively, on 17 November 2020 and 24 March 2021 at respective densities of 35 seeds/m^2^ and 50 seeds/m^2^. This difference in seed density ensures the same density of stems because WFB branch many times at the base of the plants during the winter period while SFB has one to two upright stems per plant [51]. In these conditions, we assumed that both crops are growing in two distinct environments driven by (i) the time laps in phenological development, (ii) the different weather conditions during the respective developmental stages of both crops and (iii) the different beneficial and pest communities at each crop developmental stage. No insecticides or fertilizers were applied in the trials. One fungicide treatment was applied at the bud-flowering stage to avoid the occurrence of diseases during the experiment. Chemical weeding was performed after sowing.

### 2.2. Traps, Lures and Experimental Design

Two designs of traps (Appendix B) were associated with three types of kairomonal lures, i.e., six trapping modalities were tested during field experiments. Green funnel pan traps with barrier cross bars (PV-Pherobank B.V., Wijk bij Duurstede, The Netherlands) are commonly used in the monitoring or mass-trapping of moths and beetles. These traps are composed of a transparent bucket closed by a green funnel below two perpendicular green cross bars. Semiochemical lures are placed in a cage at the center of the crossbars. The second type of trap was a white pan trap with a transparent cylinder (PB-AgriOdor, Rennes, France). This trap presents a white container surmounted by a transparent cylinder for the interception of flying insects. Semiochemical lures are deposited on a central receiver inside the cylinder. Both of these traps were filled with water containing an odorless surfactant (TRITON X-100, 0.1% *v/v*) reducing the surface tension to drown insects falling into the trap [52]. As non-lured semiochemical traps were already demonstrated to be non-attractive to BBWs [32], four repetitions of control traps, consisting of a sticky transparent surface (measuring 14.8 cm × 21 cm), were displayed in both crops (Appendix B). Semiochemical lures tested in traps included three different types of kairomones: two kairomones commercialized by AgriOdor reproducing the pod odors (AGDG) and the flower odors (AGDF). These lures were dispensed inside polyethylene vials. The third kairomone was commercialized by International Pheromone System Ltd. (Neston, UK) and reproduces the odor of flowers (IPSF). This lure consists of wax plugs that are impregnated with 1.32 g of floral kairomones combined with 0.36 g of extenders and 0.07 g of antioxidants.

The experimental site (Figure 1) covered an area of about 4 hectares and included varietal trials of field beans, lupines and peas. Traps were placed at the center of border parcels 27 m wide. The density of BBWs was assumed to be homogeneous in these parcels. Four replicates of all trap types and attractants were placed in each crop in cross combinations which means that traps were positioned in doublets associating pan and funnel traps, both lured with the same attractant. This pairing of traps aimed to avoid trapping interferences that would have been induced by different semiochemical lures within pairs of traps. Inter-trap distance in each doublet was 5 m and the doublets were spaced at a distance of 20 m from the others. Repetitions of doublets were randomly placed in both crops, a total of 24 traps were placed per crop. Traps were placed during the flower bud stage and removed at the end of crop fructification. Semiochemical attractants were renewed every two weeks. To assess the presence of insects in crops, manual catches were conducted once a week following a standard procedure. A single operator prospected a fixed pathway of 100 m length × 1 m width at the same daily period (from 15:30 to 16:30) and captured/counted adults of BBWs in the flowers or apical leaves using a truncated cone reversed over a pill box.

### 2.3. Insects Collection, Preparation and Identification

The total amount of captured insects was collected weekly from the traps and stored in ethanol 70% (*v/v*). After an initial sorting, beneficial insects belonging to Apoidea (Hymenoptera), Syrphidae (Diptera), Coccinellidae (Coleoptera) as well as pests belonging to the subfamily Bruchinae (Coleoptera: Chrysomelidae) were selected and prepared for identification. Beneficial insects were selected for their key role as biocontrol agents and/or pollinators in field bean crops. The method used for the preparation of insects followed the guidelines of Mouret et al. (2007) [53] and Fagot et al. (2022) [54]. Morphological keys of lady beetles [55,56], hoverflies [57] and wild bees [58,59,60,61] were used. Bruchid species were separated by sex and identified by J. Fagot using the key of Zampetti and Ricci [22]. Identifications were cross-checked by morphological comparison with reference collections of the Conservatory of Functional and Evolutionary Entomology (Gembloux Agro-Bio Tech, University of Liège, Belgium) [62]. 

### 2.4. Monitoring of Field Bean Phenology

As the host plant phenology and climatic parameters play an important role in BBWs population dynamics [63], climatic parameters were followed in parallel with the phenological development of WFB and SFB during the whole period of the experiment. From 19 April (i.e., the 21st week) to 2 August 2021 (i.e., the 32nd week), WFB and SFB were checked weekly to assess their developmental stages by recording the total number of nodes, the number of nodes bearing inflorescence and the number of nodes bearing pods on 20 randomly selected stems. Mean values of the nodes were used to characterize the growth and the blooming and/or fructifying stages of crops. Climatic parameters, including the maximal daily temperature (°C) and rainfall (mm) [64], were recorded by a nearby weather station. The influence exerted by the different developmental stages on the manual or semiochemical captures was qualitatively assessed by comparing the mean number of nodes with the total abundances of captured insects in each crop.

### 2.5. Statistical Analysis

Statistical analyses were performed with the software RStudio (version 2022.07.1+554) and R (version 4.1.2) [64]. First, descriptive statistics and graphs were displayed with the *ggplot2* R package [65] in order to describe (i) the population dynamics (c.f., manual and semiochemical catches) according to the two respective crop phenologies and (ii) the timing of eventual collateral effects on beneficial insects. The comparison of trapping modalities was then performed by considering winter and spring varieties together in order to provide a holistic consideration of insect communities according to the precocity of both crops. Generalized Linear Mixed-Models (GLMMs) fitted with a Poisson error distribution were used in the comparison of trapping modalities and sex ratios using the *lme4* R package [66]. The logarithmic transformations of abundances of BBWs and beneficial insects (Apoidea, Coccinelidae and Syrphidae), were used as variables to explain. Each trap was assumed statistically independent and the location of the trapping site (i.e., trap × lure; Figure 1) was specified as a random effect to minimize the pseudo-replication of the repeated sampling at each trapping site [67]. The collection date (i.e., incorporating the crop phenology), the trapping modalities, and the interaction between the trapping devices and semiochemical lures were specified as fixed effects. Multiple comparisons of means based on Tukey’s all-pairs comparisons for the trapping modalities of BBWs and beneficial insects were performed using a *multcomp* R package [68] with Bonferroni’s correction (adjusted *p*-value < 0.05). Sex ratios (count of male/count of female) were compared by ANOVA for each trapping modality.

Regarding the community analysis, abundance ranks per trapping modality were displayed using the package *BiodiversityR* [69]. Alpha diversity metrics within the captured insect dataset were characterized using Hill’s framework [70,71] at each sampling site. More particularly, species richness, Hill–Shannon and Hill–Simpson indexes were estimated by standardizing their coverage (i.e., selecting the lowest coverage value) using *iNEXT* R package [72]. Classical Shannon and Simpson indexes show linearity and replication biases which are inherent in their algebraic formulae [73]. Both these indexes are based on the relative abundance of each taxon and are influenced by the dominance-rarity pattern of the sampling. Both these indexes were mathematically transformed becoming Hill–Shannon and Hill–Simpson indexes [70] to better calculate the mean rarity of the species [71]. The coverage was used to equalize samples instead of equal effort or rarefaction standardization because this method provides a better balance of the expected species richness against the true diversity of our defined sampling system [71]. Linear Mixed-Models (LMMs) were performed to compare the generated alpha diversity metrics with the *lme4* R package [66]. As for the abundance comparisons, the trapping site was set up as a random effect. Multiple comparisons of means based on Tukey’s all-pairs comparisons for the trapping modalities of the alpha diversity metrics were performed using the *multcomp* R package [68] with Bonferroni’s correction (adjusted *p*-value < 0.05).

Last, beta species diversity was analyzed within the dataset of captured insects according to all semiochemical trapping modalities (PBAGDF, PBAGDG, PBIPSF, PVAGDG, PVAGDF and PVIPSF) using the Bray–Curtis dissimilarity matrix and Principal Coordinate Analysis (PCoA) in order to show these dissimilarities at each sampling site. Distance-based redundancy analyses (dbRDA) were then performed on the Bray–Curtis dissimilarity matrix with the *capscale* R function setting up trapping modalities and sampling sites as explanatory variables. Afterward, ANOVA with 999 permutations was performed to test the influence of trapping modalities and sampling sites on the dissimilarities of the observations. These analyses were performed using the *vegan* R package [74]. Multi-level pattern analysis (MLPA) was performed to identify the indicator species of trapping modality using the *indicspecies* R package [75]. MLPA generates the IndVal index between the species and each trapping modality and then calculates the highest association value by incorporating a correction for unequal group (i.e., trapping modality) sizes [76]. The significance of the association which is calculated with a permutation test (n = 999) was set up at 0.1.

## 3. Results

### 3.1. Host Plant Phenology and Population Dynamics of Bbws and Beneficial Insects

Traps were installed from the onset of flowering up to the ripening pods of both crops. The developmental stages of WFB and SFB are, respectively, represented in Figure 2a and Figure 3a. Climatic parameters recorded in parallel with crop phenologies are presented in Appendix C. The winter field bean crop started flowering from the 19th week (i.e., flower bud stage) but the crop development was strongly slowed down by the cold temperatures observed during the month of May (from the 19th to the 22nd weeks). The flowering of WFB increased after the 22nd week and lasted up to the 25th week with a flowering peak during the 23rd week when young pods began to form at the base of plants. The pod formation and grain filling of WFB lasted until the 31st week. Spring field bean crops started flowering during the 23rd week and lasted until the 28th week with a flowering peak observed during the 25–26th weeks. The pod formation started in the 25th week and pods were mature in the 32nd week.

Following this phenological development, twenty-four semiochemical traps were installed with four control sticky traps and were weekly collected from the 15 of April 2021 (i.e., 17th week) until the 29 July 2021 (i.e., 31st week) in WFB and from the 6 of June 2021 (i.e., 24th week) until the 2 August 2021 (i.e., 32nd week) in SFB. Manual catches/countings were also performed weekly during these periods to check the presence of BBWs in crops. The BBWs and beneficial population dynamics in WFB and SFB are, respectively, presented in Figure 2b and Figure 3b.

A total of 1380 BBWs were captured in semiochemical traps (Appendix D), 882 and 498 in WFB and SFB crops, respectively. No BBWs were captured by sticky control traps while manual catches/countings registered a total of 236 BBWs, 93 in the WFB crop and 143 in the SFB crop, respectively. A single bruchid capture was recorded in a semiochemical trap during the 19th week in WFB, but most of the adults were recorded from the 22nd week at the end of May by manual catches/countings, which coincided with the peak of the crop flowering (Figure 2b). Semiochemical traps started to capture BBWs from the 25th week which corresponds to the decreasing of flower amounts in crops when young pods began to form. This trend was also observed in SFB: manual caches/counting indicated the presence of BBWs during the peak of flowering (c.f., 25th week), while semiochemical traps captured BBWs at the end of flowering (c.f., from the 26th week) when the young pods were formed. Concerning the influence of climate, maximal daily temperatures exceeded the flying threshold of BBWs (i.e., 15 °C) from the 20th week but no semiochemical captures of BBWs were observed until the 25th week. 

A total of 1,424 beneficials were captured in semiochemical traps (Appendix A), 571 in WFB and 853 in SFB, respectively. No beneficials were captured by semiochemical sticky traps that presented other groups of insects such as Diptera species, Hemiptera species, and Ichneumonoidea wasps. No trend seemed to be highlighted concerning the field bean crops phenologies, but the temperatures and precipitations were more likely to influence the beneficial abundance in semiochemical traps (i.e., weeks presenting high temperatures and low precipitations). During the 17th to the 25th weeks, semiochemical traps captured more beneficials than BBWs in the WFB crop. In the SFB semiochemical traps, beneficials were more abundant than BBWs during the 24th to the 26th weeks. After the blooming of both crops, BBWs were more abundant in semiochemical traps than beneficials. Peaks of BBWs capture are both observed at the end of WFB and SFB blooming. 

### 3.2. Influence of Trapping Modalities on the Capture of BBWs and Beneficials 

The results of the GLMMs and Tukey’s pair comparisons with Bonferroni’s correction are provided in Figure 4. BBWs catches, Figure 4a, were significantly higher with white traps lured with floral kairomones of IPS (PBIPSF) rather than all other green trapping modalities (PVAGDF-PBIPSF z-value = −4.15; PVAGDG-PBIPSF z-value = −4.48; PVIPSF-PBIPSF z-value = −3.12; all adjusted *p*-values < 0.05). No significant differences in terms of BBWs captures were highlighted within the different associations of white or green traps with different semiochemical lures. The association of pods kairomones with the white trap (PBAGDG) captured significantly more BBWs than its equivalent green trap (PVAGDG) (z-value = −2.99; adjusted *p*-value < 0.05). Concerning the capture of beneficials, Figure 4b, the PBIPSF modality was significantly more efficient than PBAGDG and all the green trapping modalities (PBIPSF–PBAGDG z-value = 3.14; PVAGDF-PBIPSF z-value = −5.12; PVAGDG-PBIPSF z-value = −5.44; PVIPSF-PBIPSF z-value = −4.38; all adjusted *p*-values < 0.05). Both AgriOdor lures (AGDG and AGDF) were significantly more attractive in white traps than in the green traps (PVAGDF–PBAGDF z-value = −3.46; PVAGDG–PBAGDF z-value = −3.80; adjusted *p*-values < 0.05). Sex ratios of BBWs were calculated for trap capture in each trap combination (Figure 5). No statistical differences were observed. (df = 5; F-stat = 0.46; *p*-value = 0.80)

### 3.3. Analyses of Communities

All the BBWs that were captured during the experiment (c.f., semiochemical and manual traps) corresponded to the species *B. rufimanus.* Consequently, no further diversity analyses were performed on BBWs. Concerning beneficials, the community of 1424 insects captured by semiochemical traps included 20.4% of lady beetles (Coleoptera: Coccinelidae), 22.2% of hoverflies (Diptera: Syrphidae), and 57.4% of social and solitary bees (Hymenoptera: Apoidea). Bees encompassed 49 species, Coccinellidae included six species and Syrphidae included 12 species (Appendix D). *Apis mellifera* Linnaeus, 1758 (n = 321), *Bombus terrestris* (Linnaeus, 1758) (n = 275), *Coccinella septempunctata* Linnaeus, 1758 (n = 191) and *Episyrphus balteatus* (De Geer, 1776) (n = 129), were the most captured species. The abundance rankings of these dominant species according to trapping modalities are represented in Figure 6.

The comparison of alpha diversity according to each trapping modality is represented in Figure 7. At coverage-standardized samples, the richness of beneficial insects (Figure 7a) is significantly higher in PBAGDG than in the PVAGDF modality (z-value = −3.01; adjusted *p*-value < 0.05). For Hill–Shannon and Hill–Simpson alpha metrics (Figure 7b,c), the values are significantly higher in PBAGDF than in PVAGDG and PVAGDF (Shannon PBAGDF–PVAGDF z-value = −3.21; Shannon PBAGDF–PVAGDG z-value = −3.50; Simpson PBAGDF–PVAGDF z-value = −3.47; Simpson PBAGDF–PVAGDG z-value = −3.59; adjusted *p*-values < 0.05). The percentage of missing species oscillated from 34.66% to 62.22% among the trapping modalities (Table 1). Compared to the actual species richness estimated by the Chao1 index, PBAGDF would do the most collateral damage for beneficial insects, with 74.89 ± 17.44 different species captured, while the same lure combined with green traps would have the least impact on the beneficial communities, with 28.02 ± 12.92 different species captured.

Finally, the communities associated with all repetitions of trapping modalities are represented on the biplot space of the PCoA analysis in Figure 8. Overall variance explained by two principal components accounted for 29.1%. The communities were mainly structured along two principal gradients. The first gradient is the type of trap that seems strongly correlated with the first principal component that encompasses 17.4% of the explained variance. The second gradient is the type of crop (WFB or SFP) which is correlated with the second principal compound accounting for 11.7% of the explained variance. Dissimilarities of species assemblage between each observation are significantly driven by the used trapping modality (df = 5; F = 3.62; *p*-value < 0.01) and the type of crops (df = 1; F = 4.01; *p*-value < 0.02). Within these communities, MLPA analysis identified indicator 11 species that are significantly (alpha at 0.1) associated with a trapping modality, or group of trapping modalities (Table 2). The six categories incorporate six bee species (three *Bombus* spp.), three hoverfly species and two lady beetle species. 

## 4. Discussion 

A total of 48 traps were tested in two crops under hyper-infestation conditions. Data collection included insect sampling in semiochemical traps, manual catches, and phenological and climatic records during a period of fifteen weeks in one site. A total of 2804 insects of interest including 68 different species were identified. All these data were integrated to interpret major factors (biotic and abiotic information) driving population dynamics that may interfere with the traps comparison. This sampling effort successfully discriminated against the most efficient type of traps and provided a local assessment of collateral effects on selected groups of beneficials. The crop phenology was also demonstrated to manifest an influence on the attractiveness of BBWs. These local observations are discussed in the following sections.

### 4.1. Most Efficient Traps for the Capture of Bruchid, Phenological Influences on Catches and Impact on Beneficials

The trap modality that was statistically the most efficient for the capture of BBWs was the combination of IPS kairomonal lures (i.e., flowers kairomones) with a white pan trap. White traps were more effective than green traps with all semiochemical lures while previous studies suggested that green cone traps baited with flower kairomones were efficient for the capture of BBWs [32,77]. In regards to the sex ratio, IPSF lures would have been expected to present more males than females, because males were demonstrated to be more attracted by flower kairomones [32]. The AGDG lures would also have been expected to capture more females than males, as gravid females are attracted by pod kairomones for oviposition [47]. These hypotheses were not validated as more females were captured by flower kairomones on the one hand and no statistical difference could confirm the more abundant catches of females (i.e., less important sex ratio) with pod kairomones on the other hand. 

The field bean phenology played an important role in the number of BBWs that were recorded in traps. In both cultures, the number of trapped BBWs increased as the number of flowers decreased while (i) manual catches/counting attested the presence of BBWs in crops, and (ii) maximal daily temperatures were superior to the flying activity threshold of BBWs. This observation shows a strong competition of the crop odor with the attractiveness of the semiochemical traps that decrease their efficiency, as suggested by previous studies [41]. The most important capture of BBWs in WSB and SFB crops (more than 500 BBWs) occurred during the 30th week which corresponds to the end of SFB flowering which is supposed to trigger the departure of adults from both field bean crops in search of a food source [28,29,38,42] and may drastically increase the attractiveness of semiochemical lures. Manual catches/countings were maximal during the peak of flowering and continued until the end of blooming before dropping sharply. This can be explained by two factors. First, the peak of crop flowering is reported as the period of highest BBW density [39]. Secondly, the concentration for feeding and mating occurs in the flowers which are located on the upper parts of the plants, the individuals found there are easily visible by the operator performing the manual catches/countings. After flowering, the males leave the crop and the females spread into the crop to lay eggs at the base of plants and became difficult to be caught/counted.

Beneficials considered in this study focused on bees, hoverflies and lady beetles. Populations were not as dependent on crop phenologies as BBWs because they were captured by semiochemical traps during the whole experimental period. An increase in captures is nevertheless observed during the flowering periods of field bean crops. This is consistent with other studies stating that crop flowering is a good predictor of the beneficials abundance [78]. Climatic factors also influenced captures, i.e., warm and low precipitation which is common for poikilothermic flying organisms [79]. Respectively, 312 and 415 bycatches were observed in WFB and SFB before the BBWs captures exceeded the captures of beneficials. This constitutes an important impact on auxiliaries before the BBWs infestation. The impacts of trapping modalities are contrasted in terms of abundance and diversity. PBIPSF was the most effective trap regarding the abundance of beneficials, but PBAGDF was the most effective one on species richness. Thus, the combination of white traps and floral attractants had the most important effect on beneficial communities.

### 4.2. Analyses of Communities of BBWs and Beneficials: Balanced Impacts According to Functional Groups and Their Ecology

Analyses of the BBWs community revealed that only one population of *B. rufimanus* was present in field bean crops. None of the four other BBWs recorded in Belgium that develop in faba bean seeds were observed [21,22]. More particularly, no *B. pisorum,* an important pest of pea that may use seeds of *V. faba* as larval host [22,80], was captured in semiochemical traps despite the presence of pea parcels close to the trials. This may suggest that the semiochemical lures established on the basis of electroantennographic studies on *B. rufimanus* are specific to this pest. Other BBWs species could be more sensitive to other specific VOCs or to a different ratio in ubiquitous volatiles emission such as (Z)-3-hexenyl acetate, limonene, caryophyllene, linalool, myrcene [81,82]. 

The community of beneficials (bees, hoverflies and lady beetles) observed in semiochemical traps included 67 species belonging to 31 different genera and eight families (see Appendix D). The most abundant species were *A. mellifera*, *B. terrestris*, *C. septempunctata and E. balteatus*, which are commonly observed in field bean crops [83,84,85]. They include three functional groups according to the ecology of different species: (i) pollinators (bees and hoverflies), (ii) aphid predators (lady beetles and hoverflies) and (iii) pollinators and aphid predators (hoverflies). Ecological differences within and between these groups induce different vulnerabilities of the populations to semiochemical traps that must be considered in the assessment of their impact on ecosystem services. The pollinator community is important for crop yield as cross-pollination increases the number of pods per plant, the number of seeds per pod, and seed weight, which can increase total yield by up to 185% compared to self-pollination [78]. However, not all species contribute equally to cross-pollination. Among dominant species of pollinators, it was reported that bumblebees contributed more efficiently to the fertilization of bean flowers (i.e., the number of pods was significantly higher) in front of honey bees and hoverflies [84]. Additionally, temporal variation in population dynamics is a key determinant of semiochemical trap impacts on pollinator communities. Indeed, the annual colonies of some bumblebees develop from founding queens whose flight periods cover the months of March, April and May. Therefore, the abundance of *Bombus* spp. captured during these periods underestimate the impact on their population because they represent potential colonies of 100 to 400 individuals [85]. It should be noted that not all the *Bombus* species present the same pollination efficiency, some being nectar robbers such as *B. terrestris* workers [86] while others being cuckoos of other bumblebee species such as *B. vestalis* [87].

Unfortunately, the semiochemical traps captured bee species that are exposed to extinction issues in Belgium which could be one of the most detrimental impacts of this IPM alternative. *Andrena ovatula* (Kirby, 1802), *A. wilkella* (Kirby, 1802), *Bombus hortorum* (Linnaeus, 1761) (indicator species in IPSF traps), *B. vestalis* (Geoffroy, 1785) are near threatened species. *Lasioglossum minutulum* (Schenck, 1853) and *Stelis signata* (Latreille, 1809) are classified as vulnerable in Belgium and *Nomada fuscicornis* Nylander, 1848 (the cuckoo bee of *Panurgus calcaratus* (Scopoli, 1763) which was not present in our sampling) is endangered [88]. Semiochemical traps are also damaging for oligolectic bee species such as *Chelostoma campanularum* (Kirby, 1802), *Megachile ericetorum* Lepeletier, 1841 and *Melitta leporina* (Panzer, 1799). Only *M. ericetorum* strictly forages in Fabaceae family, mainly *Lotus* spp. and *Lathyrus* spp. flowers [89]. None of the captured hoverfly species are exposed to a European extinction threat [90] Most of the species show aphidophagous behavior at the larval stage except for *Eristalis tenax* (Linnaeus, 1758) and *Xylota lenta* Meigen, 1822 whose larvae show microphagous and saproxylic behavior, respectively [91].

Concerning aphid predators, *E. balteatus* and *C. semptempunctata* were the most abundant taxa in all trapping modalities. These two species are reported to be the most abundant aphidophagous in the agrosystems of Europe and Belgium [92]. *E. balteatus* larvae and adults have different diets. Adults feed on pollen and nectar while larvae feed on aphids. *Coccinella semptempunctata* is a polyphagous predator, both larvae and adults are known to feed on Aphidoidea, Psylloidea, Coccoidea and mites [93]. Larvae of *E. balteatus* may consume up to 396 aphids in field conditions, and up to 1322 under laboratory conditions [94], while the respective consumption of *C. semptempunctata* larvae (third instar) and female adults are, respectively, 277 or 204 aphids per day [95]. No study clearly compared the predation rate of lady beetle and hoverfly larvae, and their consumption rate depends on many factors such as temperature, host plant, aphid species and experimental conditions. Ecological factors such as prey densities and intra-guild predation may also impact the co-existence of these aphidophagous species [96]. Both of these two species should remain the least impacted by semiochemical traps. Our results showed that hoverflies were more impacted by white traps and that lady beetles by green traps, as reported by other studies [97,98,99].

### 4.3. Maximising B. Rufimanus Trapping and Minimising Beneficial Insect Trapping: A Dilemma

Semiochemical traps offer a sustainable pest management tool that can contribute to the promotion of grain legumes for European crop diversification, but this tool should not impact the ecological services of field bean crops. In this study, the ratio of captured beneficials per captured BBWs was nearly higher than 1, which emphasizes that these semiochemical traps present a similar impact on beneficials and BBWs. This excludes, therefore, a potential mass trapping strategy against BBWs that spares collateral effects on organisms of agronomic interest. To mitigate the capture of beneficial insects, trap design could be improved and their spatiotemporal deployment could be adapted to the influence of flowering on the capture of BBWs. 

Adaptation measures that could improve the selectivity of semiochemical traps and their implementation in the field are multiple. Firstly, other types of traps can be used instead of water-containing traps. These traps are subjected to drying in sunny field conditions which imply repeated field interventions to maintain the trapping devices which are costly in time and energy for growers. Sticky white delta traps are currently being studied in alternative to white pan traps (Ené Leppik, personal communication). Then, the trap selectivity can be increased by (i) the placement of a grid preventing larger insects to reach the trap killing agent [49], (ii) the improvement of semiochemical attractiveness (improved or new semiochemical) [29], and (iii) the displaying of a more selective killing agent in trapping devices, such as entomopathogenic fungi or double-stranded RNA [29,100]. The reduction in trapping periods by displaying traps at the appropriate moment, i.e., from the flowering peak, would also limit the captures of beneficials. Finally, an integrated pest management strategy that can be considered is the trap crop approach. This method takes advantage of the effectiveness of semiochemical traps and the influence of crop flowering. It consists in attracting BBWs in an early flowering cultivar and capture them with semiochemical traps to protect late flowering cultivar crops [101]. 

## 5. Conclusions

The multiple advantages of introducing leguminous plants in crop rotations are well documented in the literature. However, European farmers rarely opt for their introduction in cropping systems which is mainly due to biotic sensitivities causing irregular and uncertain harvests in terms of quality and quantity. Research is needed in the development of better-performing cultivars, innovative and efficient technical itineraries for the sustainable control of pests and the producing food of a quality that could be valorized in national markets. In this study, a potential new biocontrol strategy was investigated by testing and comparing semiochemical traps for the capture of BBWs as well as their collateral effects on beneficial insects in two field bean crops (i.e., winter field bean and spring field bean). Key factors of the agrosystem influencing BBWs population dynamic (i.e., the phenology and the climate) were followed in parallel to interpret the evolution of captured pests. The best-suited traps for the capture of BBWs were identified as well as their impact on beneficial communities. The strong influence exerted by the flowering of crops on the capture of BBWs was described and a complete description of captured communities is provided. Recommendations and guidance are suggested on this background for the development of optimal biocontrol strategies that minimize the impact on beneficial insects. These results support the development of new control methods in legume crops which are essential crops for the diversification of European cropping rotations and for the resilience of the food production system.

## Figures and Tables

**Figure 1 insects-14-00153-f001:**
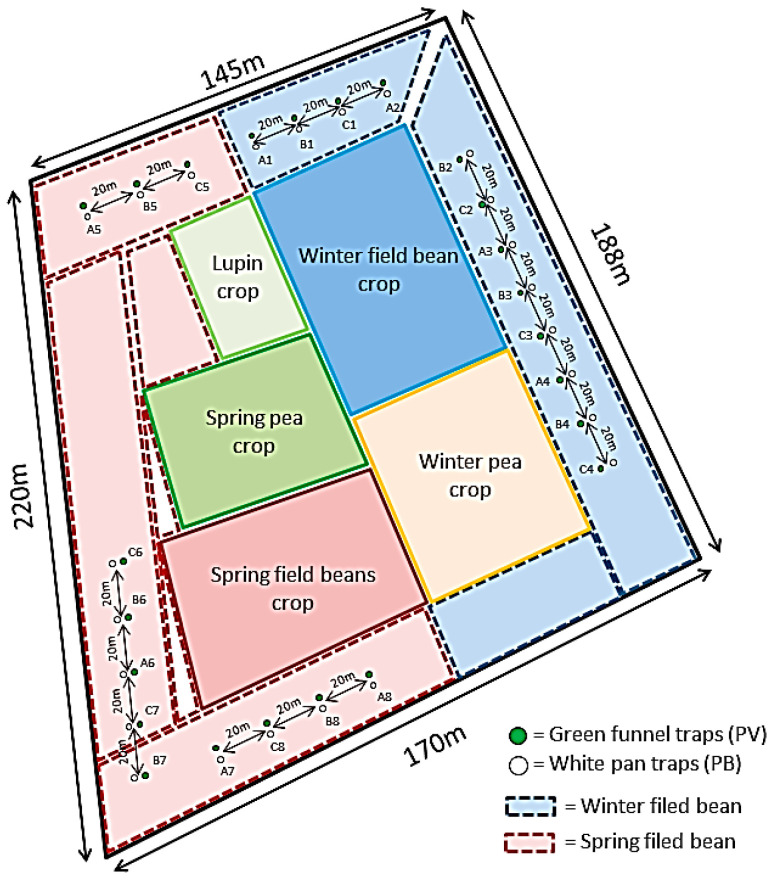
Experimental site and trap disposition in winter (WFB) and spring field bean (SFB) parcels. Each lure is set in doublets of each type of trap (PV and PB). A1 to A8: replicates of pod kairomones from AgriOdor (AgdG) in winter and spring field beans; B1 to B8: replicates of flower kairomones from AgriOdor (AgdF) in winter and spring field beans; C1 to C8: replicates of flowers kairomones from International Pheromone System (IPSF) in winter and spring field beans.

**Figure 2 insects-14-00153-f002:**
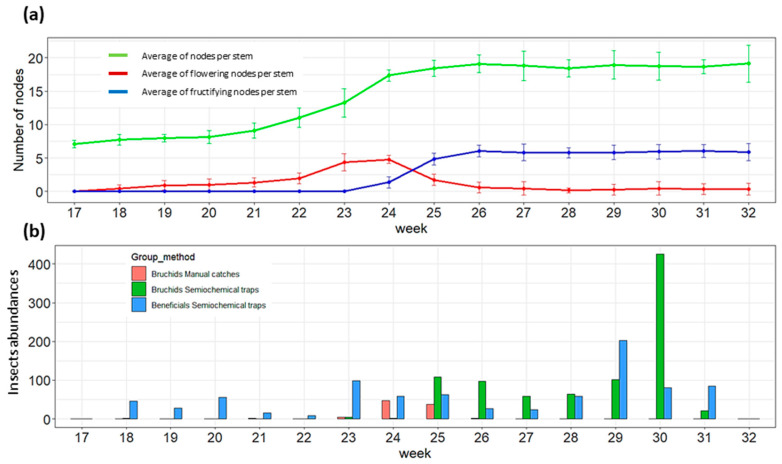
Population dynamics of broad bean weevils (BBWs) and beneficial insects (Apoidea, Syrphidae and Coccinellidae) in WFB according to the crop development. (**a**) Development of WFB: mean number of nodes per stem (+/− standard deviation-green), mean number of flowering nodes per stem (+/− standard deviation-red) and mean number of fructifying nodes per stem (+/− standard deviation-blue). (**b**) The temporal abundance of BBWs and beneficial insects according to trapping method: manual trapping (red), captured-semiochemical BBWs (green) and captured-semiochemical beneficials (blue).

**Figure 3 insects-14-00153-f003:**
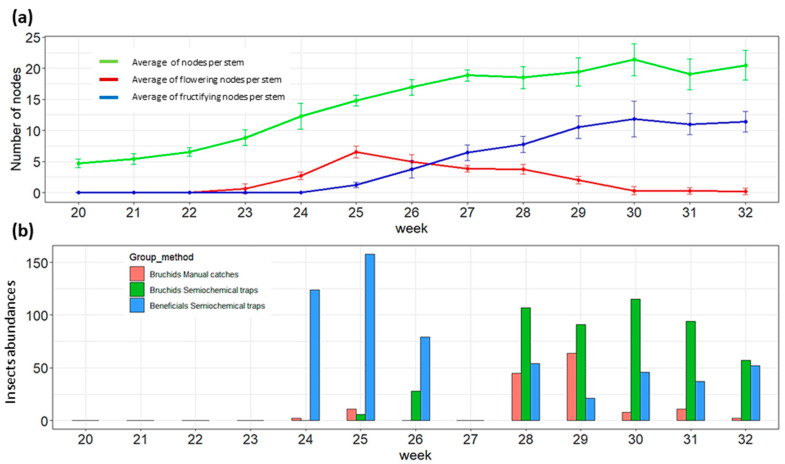
Population dynamics of broad bean weevils (BBWs) and beneficial insects (Apoidea, Syrphidae and Coccinellidae) in SFB according to the crop development. SFB according to crop development. (**a**) Development of WFB: mean number of nodes per stem (+/− standard deviation-green), mean number of flowering nodes per stem (+/− standard deviation-red) and mean number of fructifying nodes per stem (+/− standard deviation-blue). (**b**) The temporal abundance of BBWs and beneficial insects according to trapping method: manual trapping (red), captured-semiochemical BBWs (green) and captured-semiochemical beneficials (blue).

**Figure 4 insects-14-00153-f004:**
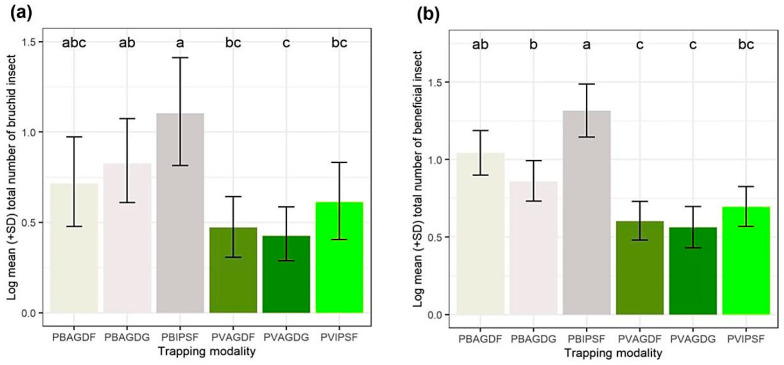
Log of the mean number (± SD) of captured insects per trapping modality. (**a**) Broad bean weevils (BBWs); (**b**) beneficial insects. In (**a**,**b**) graphics, the shades of grey correspond to the white traps and the shades of green correspond to the green traps. Log of the means with different letters are significantly different (Bonferroni’s adjusted *p*-value < 0.05).

**Figure 5 insects-14-00153-f005:**
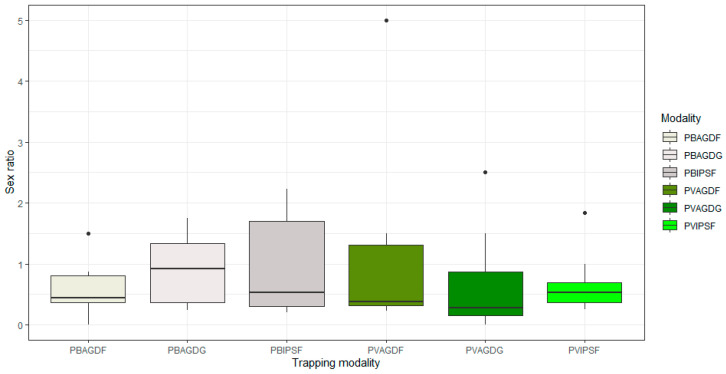
Boxplots of sex-ratios led to BBWs captures according to trapping modalities. The black dots are outliers, the bold horizontal line in each boxplot corresponds to the median of sex-ratio between the upper and lower quartile (i.e., the interquartile range or the middle 50% scores) and the upper and lower whiskers (i.e., the score outside the middle 50%).

**Figure 6 insects-14-00153-f006:**
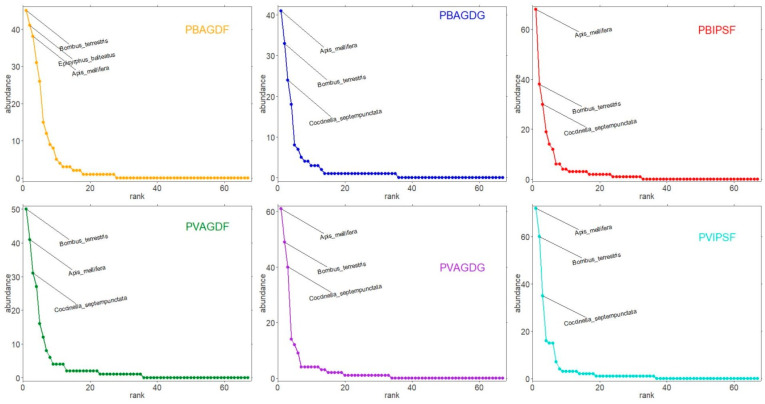
Abundance rank analysis. The top three species are labeled in each graphic for each trapping modality: orange curve for PBAGDF; blue curve for PBAGDG; red curve for PBIPSF; green curve for PVAGDF; purple curve for PVAGDG; turquoise curve for PVIPSF.

**Figure 7 insects-14-00153-f007:**
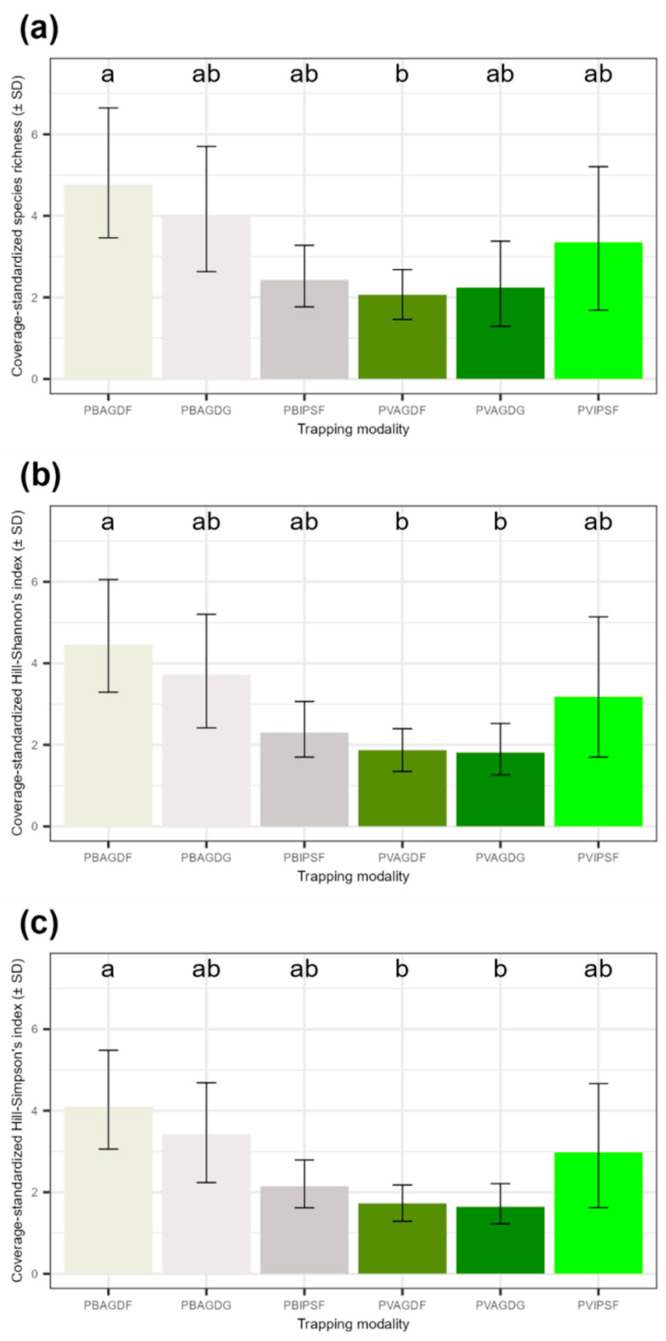
Alpha diversity metric comparisons per trapping modality for beneficial insects: (**a**) coverage-standardized species richness, (**b**) Hill–Shannon index and (**c**) Hill–Simpson index. In (**a**–**c**) graphics, the shades of grey correspond to the white traps and the shades of green correspond to the green traps. Alpha diversity means with different letters are significantly different (Bonferroni’s adjusted *p*-value < 0.05).

**Figure 8 insects-14-00153-f008:**
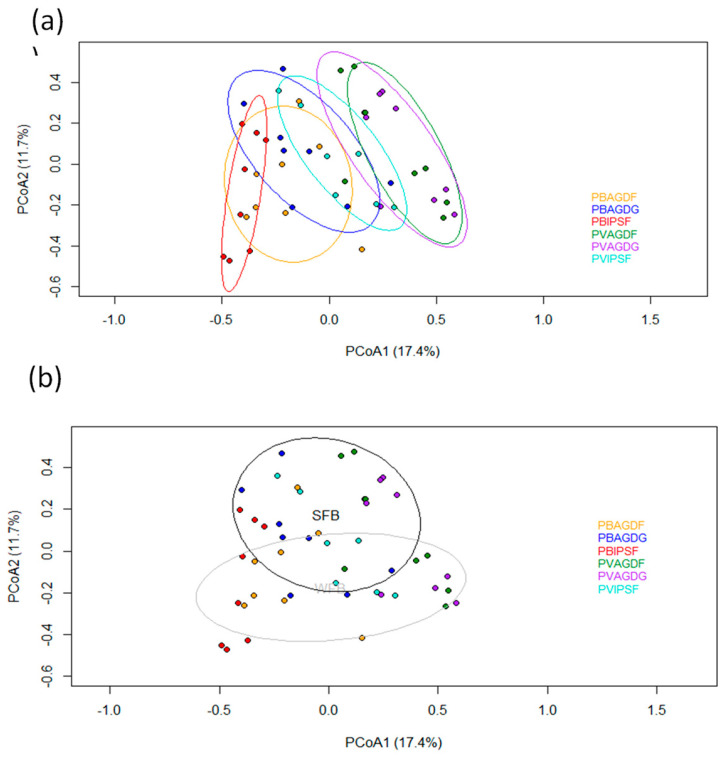
Principal Coordinates Analysis (PCoA) of trapping modalities at each sampling sites. Each dot color corresponds to a trapping modality (i.e., lure x trap): red = PBIPSF; orange = PBAGDF; blue = PBAGDG; turquoise = PVIPSF; purple = PVAGDG; green = PVAGDF. (**a**) Ellipses show the 75% confidence interval of the locations grouped by trapping modality with the same correspondence color of the dots. (**b**) Ellipses show the 75% confidence interval of the locations grouped by Winter Field Bean culture (WFB) in grey and Spring Field Bean culture (SFB) in black.

**Table 1 insects-14-00153-t001:** Species richness and chao1 index per trapping modality.

Trapping Modality	Species Richness	Chao1 Index ± SE	Potential Estimation of Missed Taxa Proportion [%]
All the experiment	67	114.88 ± 28.50	41.68
PBAGDF	45	74.89 ± 17.44	39.91
PBAGDG	34	52.03 ± 12.07	34.66
PBIPSF	37	68.62 ± 23.06	46.08
PVAGDF	16	28.02 ± 12.92	42.90
PVAGDG	18	47.65 ± 28.08	62.22
PVIPSF	24	38.88 ± 12.28	38.28

**Table 2 insects-14-00153-t002:** Indicator species of captured communities in function of trapping modalities. ‘*’ indicates *p*-value < 0.05.

Trapping Modality	Taxa	Indicator Statistic	*p*-Value
PBAGDF	*Lasioglossum minutissimum*	0.48	0.07
	*Andrena subopaca*	0.44	0.05 *
PBIPSF	*Apis mellifera*	0.76	>0.01 *
	*Melanostoma mellinum*	0.54	>0.01 *
	*Lasioglossum laticeps*	0.44	0.08
PVAGDF + PVAGDG	*Harmonia axyridis*	0.44	0.02 *
	*Coccinella septempunctata*	0.43	0.04 *
PBAGDF + PBIPSF	*Episyrphus balteatus*	0.59	>0.01 *
	*Spaerophoria scripta*	0.53	>0.01 *
PBIPSF + PVIPSF	*Bombus hortorum*	0.60	>0.01 *
PBAGDF + PBIPSF + PVIPSF	*Bombus pascuorum*	0.48	>0.01 *

## Data Availability

Insects collections are stored at the Conservatoire du laboratoire d’Entomologie Fonctionnelle et Evolutive–Gembloux Agro-Bio Tech (Université de Liège). Available online: https://www.gembloux.ulg.ac.be/entomologie-fonctionnelle-et-evolutive/presentation-du-conservatoire/ (accessed on 1 February 2022).

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
