# Peer review of "Impacts of Semiochemical Traps Designed for Bruchus rufimanus Boheman 1833 (Coleoptera: Chrysomelidae) on Nontarget Beneficial Entomofauna in Field Bean Crops"

_insects, 2023, doi:10.3390/insects14020153_

Round 1

Reviewer 1 Report

I have a few comments on the manuscript.

- The introduction is very large. It is necessary to shorten the introduction and leave the most important text.

- Write down the purpose and objectives of the research more clearly.

- Line 206. What does "beneficial insects" mean? Why haven't other "not beneficial" insects been studied?

- In Conclusion, references to literature are not used.

- Do the authors have data on the average occurrence of insects in traps?

- It seems to me that many more insects should fall into the traps than the authors indicated in the appendix. If my assumption is correct, then the list is incomplete. If the authors indicated all the insects, then the question arises: why did so few insects get into the traps? What is the reason for such a small biodiversity? Have butterflies and other species of beetles been caught?

- Appendix C. Write completely Latin names with the authors of the first descriptions, using modern insect taxonomy.

Author Response

Dear Editors, dear Reviewers,

First of all, we would like to thank you for your proofreading and for all your constructive remarks and suggestions which have reoriented the writing of this manuscript towards the most relevant elements with regard to the problematic addressed.

As you will notice in this new manuscript, answers provided to reviewers’ suggestions have induced the suppression of irrelevant and long information that were not justified in the introduction. The manuscript was written again by fitting better sentences structure, restructuring the development of the information, and a revision of English has been performed according to the received corrections. New informative elements and references have also been included for a more comprehensive reading and some paragraphs were added in the discussion to integrate reviewer’s remarks. All changes made to the original text have been highlighted in blue using the “Track Changes” function in the new manuscript as requested by Editors.

We hope that all your remarks could correctly be responded to, and that your expectations have been met in this new manuscript. We remain available to answer your suggestions regarding to these modifications. The application of each remark is point to point explained here below.

Thank you for considering it for review. We appreciate your time and look forward to hearing from you at your earliest convenience.

Arnaud Segers

Reviewer 2 Report

Please see attached review.

Author Response

(The authors gave the same response as above.)

Round 2

Reviewer 1 Report

Dear authors. Thank you for your answers.

Author Response

Dear Reviewers, dear Editors,

Authors would like to thank you for your proofreading and for all your constructive remarks and suggestions received with these minor revisions. As you will notice, we applied all your suggestions in the new version of the manuscript. Major revisions (c.f., previous version of the manuscript) were all applied in order to point the minor revisions that are highlighted in blue using the “Track Changes” function as requested.

The application of each remark is point to point explained in attached file. We hope that your expectations have been met in this new manuscript. We remain available to answer your suggestions regarding to these modifications.

Thank you for considering it for review. We appreciate your time and look forward to hearing from you at your earliest convenience.

Arnaud Segers

Reviewer 2 Report

Within the confines of the study, the authors have responded to most of my original concerns.  I still think that only 1 year and 1 study site is a very limited sample, from which we cannot gain too much information.  It also makes no sense to me that they would pair the different trap types and not include that in the analysis...what is the point of pairing? English editing is still needed on this version of the manuscript.

Author Response

Dear Reviewers, dear Editors,

Authors would like to thank you for your proofreading and for all your constructive remarks and suggestions received with these minor revisions. As you will notice, we applied all your suggestions in the new version of the manuscript. Major revisions (c.f., previous version of the manuscript) were all applied in order to point the minor revisions that are highlighted in blue using the “Track Changes” function as requested.

The application of each remark is point to point explained in attached files. We hope that your expectations have been met in this new manuscript. We remain available to answer your suggestions regarding to these modifications.

Thank you for considering it for review. We appreciate your time and look forward to hearing from you at your earliest convenience.

Arnaud Segers
